# Nuclear Organization during Hepatogenesis in Zebrafish Requires Uhrf1

**DOI:** 10.3390/genes12071081

**Published:** 2021-07-16

**Authors:** Bhavani P. Madakashira, Chi Zhang, Filippo Macchi, Elena Magnani, Kirsten C. Sadler

**Affiliations:** Program in Biology, New York University Abu Dhabi, Abu Dhabi 129188, United Arab Emirates; bm116@nyu.edu (B.P.M.); cz21@nyu.edu (C.Z.); fm1442@nyu.edu (F.M.); em188@nyu.edu (E.M.)

**Keywords:** liver development, organogenesis, nuclear organization, nuclear structure, *uhrf1*, DNA methylation, zebrafish

## Abstract

Acquisition of cellular fate during development is initiated and maintained by well-coordinated patterns of gene expression that are dictated by the epigenetic landscape and genome organization in the nucleus. While the epigenetic marks that mediate developmental gene expression patterns during organogenesis have been well studied, less is known about how epigenetic marks influence nuclear organization during development. This study examines the relationship between nuclear structure, chromatin accessibility, DNA methylation, and gene expression during hepatic outgrowth in zebrafish larvae. We investigate the relationship between these features using mutants that lack DNA methylation. Hepatocyte nuclear morphology was established coincident with hepatocyte differentiation at 80 h post-fertilization (hpf), and nuclear shape and size continued to change until the conclusion of outgrowth and morphogenesis at 120 hpf. Integrating ATAC-Seq analysis with DNA methylation profiling of zebrafish livers at 120 hpf showed that closed and highly methylated chromatin occupies most transposable elements and that open chromatin correlated with gene expression. DNA hypomethylation, due to mutation of genes encoding ubiquitin-like, containing PHD and RING Finger Domains 1 (*uhrf1*) and DNA methyltransferase (*dnmt1*), did not block hepatocyte differentiation, but had dramatic effects on nuclear organization. Hepatocytes in *uhrf1* mutants have large, deformed nuclei with multiple nucleoli, downregulation of nucleolar genes, and a complete lack of the nuclear lamina. Loss of lamin B2 staining was phenocopied by *dnmt1* mutation. Together, these data show that hepatocyte nuclear morphogenesis coincides with organ morphogenesis and outgrowth, and that DNA methylation directs chromatin organization, and, in turn, hepatocyte nuclear shape and size during liver development.

## 1. Introduction

The size, shape, and distribution of landmark structures in the nucleus are defining features of differentiated cells. These features are the architectural foundation for genome organization, which is directly related to the gene expression profiles that drive cellular functions. Thus, the precise control of gene expression patterns that regulate cell-specification, differentiation, and maintenance of cell fate during development are directly influenced by their accessibility [1,2,3,4]. The dynamic changes in genome organization during development play a role in fate acquisition and are also important for retaining the ability to change fate in response to injury or other differentiation signals [5,6]. How these features are patterned during organogenesis in whole animals is not well understood.

The nuclear organization is dictated by structural forces and epigenetic marks, identified through morphological features and specific histone and DNA modifications [7]. Landmarks of the repressive epigenome are heterochromatin foci, nuclear lamina, and the nucleolar rim, all of which are marked by 5 methylcytosine (5MeC) and histone H3 lysine 9 di- or tri-methylation (H3K9me2/3) [8,9]. These heterochromatin marks serve to compact the chromatin, making it inaccessible to transcription factors, impeding the process of DNA replication, and retaining large regions of the genome in structures that maintain the structural integrity of the nucleus [7,10]. Importantly, DNA in these regions are largely transcriptionally inactive. During differentiation, genes that are not relevant to the function of the cell type are locked away in the lamina or other regions of heterochromatin. In addition, most cell types silence transposable elements (TEs) by packaging them in heterochromatin [11]. Thus, the formation of distinct nuclear structures by deposition of repressive epigenetic marks is an essential feature of cell morphology and function.

Undifferentiated pluripotent cells are characterized by spherical nuclei with predominantly euchromatic features [12]. Similarly, stem cells have fewer heterochromatin foci and are less organized than differentiated cells [13,14], showing that pluripotent cells are more euchromatic and allow transcription factor access to genes for multiple lineages [15]. This pattern is also observed in early embryos, where the nucleus is largely euchromatic and develops structural landmarks during gastrulation and somitogenesis [11,16,17]. Defining when these patterns of genome organization and nuclear structure develop and how these patterns are set is important for understanding the structure-function relationships in gene regulation and nuclear organization that are critical for developmental events.

Deposition of the heterochromatin marks that define distinct regions of differentiated cell nuclei is carried out by several chromatin-modifying complexes. Ubiquitin-like, containing PHD and RING finger domains 1 (Uhrf1) is a well-conserved epigenetic regulator [18], which is essential for maintenance DNA methylation [19,20,21]. In addition, Uhrf1 serves as both a reader of the histone code, as a writer of this code by ubiquitinating multiple histone H3 sites [22,23,24,25], and also by recruiting modifying enzymes, including DNA methyltransferase 1 (Dnmt1) [21,26,27], and the H3K9 methyltransferase G9 [28]. We [29,30,31,32,33,34] and others [21,22,23,28,35,36] have identified defects in DNA replication, mitosis, cell survival in cells lacking Uhrf1, as well as gross morphological changes in the pre-malignant nuclei of cells that overexpress UHRF1 [37]. This suggests that Uhrf1 regulates nuclear organization either through direct interaction with the structural components of the nucleus or indirectly through its role in mediating DNA methylation and other heterochromatin marks.

The mechanisms of liver development have been extensively investigated [38], and studies in zebrafish have been important for understanding many fundamental aspects of liver specification, hepatic differentiation, and outgrowth [39,40]. There is limited information about the cellular and nuclear changes in developing hepatocytes, and advancing this is important as it has been well established that distinct nuclear structure, nuclear cytoplasmic ratio, and nuclear shape and size are all distinctive for unique cell types. In the liver, progenitor cells are distinguished by the high nuclear: cytoplasmic ratio, and the relatively small size and ellipsoid shape compared to mature hepatocytes characterized by large, spherical nuclei with prominent nucleoli and nuclear lamina [41,42]. This contrasts with the small biliary nuclei, which have large prominent patches of heterochromatin [43,44,45]. These features change dramatically during malignant transformation and the prominent nucleoli, small and irregular nuclei in liver cancer, are pathognomonic for the stage of these tumors [46]. Indeed, the structural features of the nucleus are directly related to cell function, as the size and number of nucleoli are linked to transcriptional activity, which, in turn, depends on the rate of cell growth and proliferation [47].

Hepatocytes and biliary cells carry out the majority of liver functions. These cells arise from a common progenitor, and during development, undergo differentiation, proliferation, and morphological changes culminating in the formation of a complex organ that carries out diverse functions, including serum protein secretion, xenobiotic metabolism, bile secretion, and transport and energy metabolism [38]. At the same time, the intracellular structures in these cells undergo dramatic changes to both shape and hepatic architecture to allow the cells to carry out these important functions. However, little is known about how these changes are coordinated and what their functional impact is on the development and function of the liver. In particular, there are clear structural features of the hepatocyte nucleus that are relevant to function–such as a large population of ribosomes necessary for the immense requirements for protein synthesis to generate the serum proteins and metabolic enzymes essential for hepatic function. Cellular structure and functional changes that accompany hepatocyte differentiation, therefore, can both be used as a marker of differentiation and acquisition of hepatic function and also likely reflect the functional maturation of these cells.

Zebrafish provide an excellent model to study hepatogenesis [39,40], and many of the genes and processes that are known to regulate hepatic specification and outgrowth have been discovered in this model. In zebrafish, liver specification occurs during the first two days post-fertilization (dpf), and markers of hepatocyte differentiation are observed as early as 48 h post-fertilization (hpf). The hepatic bud is first visible between 60–72 h post-fertilization (hpf) when hepatocytes are identified as differentiated, and hepatoctye maturation and proliferation during hepatic outgrowth and morphogenesis occur simultaneously [48]. By 120 hpf, hepatic outgrowth is complete and the liver has undergone morphogenesis, and most functions of the liver have matured. The rapid developmental timeline, the widespread use of cutting-edge genetic and imaging approaches, and the ease of generating a large number of synchronously developing embryos makes this system highly advantageous for studying development. Moreover, many models of liver disease have been developed in zebrafish larvae [49,50], extending the relevance of findings in this system. With the advent of techniques allowing for genome-scale profiling using small sample sizes, genome-wide profiling of the chromatin landscape is now possible for tissues from zebrafish embryos, and a comprehensive chromatin map of multiple adult zebrafish tissues was recently reported [51]. The advantages of coupling cutting-edge imaging with genomics in a vertebrate model make this a powerful system to address questions related to cell biology, gene regulation, and development.

In this study, we tested the hypothesis that hepatocyte maturation during development was characterized by acquisition of distinct nuclear morphology and a chromatin landscape dictated, in part, by the pattern of DNA methylation. Using the morphological assessment of the nuclear lamina and nucleolus coupled with chromatin accessibility, DNA methylation, and transcriptomics, we assessed how the structure and key landmark features of the hepatocyte nuclei change during zebrafish hepatic outgrowth. Our previous studies showed that *uhrf1* mutant zebrafish undergo hepatic specification and differentiation, but fail to undergo outgrowth and morphogenesis, due to cell cycle arrest and apoptosis [31,52], accompanied by loss of DNA methylation [29,31,33]. This study shows that *uhrf1* mutant hepatocytes have large, disorganized nuclei that lack nuclear lamina and have multiple nucleoli, features that are phenocopied by *dnmt1* mutation. This suggests that DNA methylation is required for establishing hepatocyte nuclear structure during liver maturation.

## 2. Materials and Methods

### 2.1. Zebrafish Husbandry and Genotyping

Adult zebrafish were maintained in a circulating aquaculture system on a 14:10 h light: dark cycle at 28 °C. The *uhrf1*^hi272^ allele [53] was maintained by genotyping adults to identify heterozygous carriers as described [53]. Homozygous mutant embryos were generated by crossing heterozygous adults and were identified based on distinctive phenotypes as described [31,52,54] or by genotyping individual embryos as described [52]. *uhrf1*^hi272+/−^ adults were genotyped by PCR as described [52]. *Tg(fabp10a: nls-mcherry)* in *uhrf1*^hi272+/−^ background was used for all experiments for ease of liver detection [37]. The *Tg(fabp10a: CAAX-EGFP)* line (allele *mss8Tg*) used to generate liver-specific ATAC-Seq libraries, as previously described [55]. Homozygous *uhrf1^hi272^*^−/−^ mutant larvae are hereafter called *uhrf1^−/−^*. All protocols were approved by the NYU Abu Dhabi Institutional Animal Care and Use Committee (IACUC).

### 2.2. RNA and DNA Extraction

For each sample, 10 to 20 livers were microdissected from *Tg(fabp10a: CAAX-EGFP)* larvae to facilitate liver identification according to published protocols [56]. RNA was extracted using Trizol (15596026, Invitrogen, Waltham, MA, USA) following the manufacturer’s instructions with modifications. Briefly, during precipitation in isopropanol, 10 μg of Glycoblue (4486802, Thermo Fisher Scientific, Waltham, MA, USA) was added, and precipitation performed overnight at −20 °C followed by 1 h centrifugation at 12,000× *g* at 4 °C. The resultant RNA pellet was resuspended in water and quantified by Qubit RNA High Sensitivity kit (4483369, UTECH). For genomic DNA extraction, 40 livers were subjected to DNA extraction buffer (10 mM Tris-HCl pH9, 10 mM EDTA, 200 mM NaCl, 0.5% SDS, 200 μg/mL proteinase K) followed by DNA precipitation with isopropanol as previously described [29]. DNA was resuspended in water and quantified by Qubit dsDNA High Sensitivity kit (4483365, UTECH).

### 2.3. cDNA Production and qPCR

Three hundred nanograms of RNA extracted from microdissected livers was retrotranscribed to cDNA using Qscript cDNA synthesis kit (101414-096, Quanta Bio, Beverly, MA, USA) following the manufacturer’s instructions. cDNA was diluted to 1 ng/μL, and 5 μL was used per reaction for qPCR using Maxima R SYBR green/ROX master mix (S7563, Thermo Fisher Scientific). *rplp0* was used to normalize expression levels by using the calculations for delta-Ct, and WT siblings were used to calculate delta-delta-Ct (DDCt) as previously described [57]. To determine changes in expression between control and experimental samples, the fold change was calculated and expressed as the log 2 value (L2FC) for display. All experiments were performed on samples from at least three independent clutches, with the number of biological replicates indicated for each experiment. Primer information is provided in Appendix A.

### 2.4. RNA-Seq

We microdissected livers from 2 and 3 clutches of embryos at 78 hpf and 120 hpf, respectively. For the 78 hpf dataset, 100 ng of RNA was used for library preparation according to the manufacturer’s instructions (Illumina, San Diego, CA, USA). Libraries were sequenced on NextSeq 550 (Illumina) to obtain 150 bp paired-end reads. Raw Fastq files quality was assessed by using FASTQC (http://www.bioinformatics.babraham.ac.uk/projects/fastqc, accessed on 10 March 2021), and the reads were quality trimmed using Trimmomatic [58] to remove low quality reads and adapters. Qualified reads were mapped to the reference genome Danio rerio GRCz10 by HISAT2 [59] with default parameters. To estimate and compare gene expression in different data sets, raw reads were counted with HTseq [60] with union mode. Statistical analysis was performed with DESeq2 [61], while the raw reads count from two time points was adjusted for batch effect with Combat-seq [62]. The 120 hpf datasets were previously published [63]. Datasets are available in GEO (GSE130800 and GSE104953).

### 2.5. ATAC-Seq

Livers from 120 hpf zebrafish larvae of *Tg(fabp10a:CAAX-EGFP)* [55] were microdissected from 10 larvae, incubated in 2.5% Trypsin in PBS for 10 min at 37 °C followed by pelleting and resuspension in PBS to yield single-cell suspension. Single cells were assessed for cell viability and cell number with a Hemocytometer under a stereo microscope. Cell suspensions with greater than 94% viability were used for ATAC-seq library generation as described [64] with minor modifications. Libraries were prepared from 3 biological replicates. Prior to transposition, 10,000 healthy cells in suspension were subjected to centrifugation (500× *g*, 5 min, 4 °C) to remove debris and dead cells. The transposition reaction was carried out in agitation with 0.5 μL Tagment DNA Enzyme 1 (TDE1, FC-121-1030, Alliance Global, Dubai, United Arab Emirates) in a total volume of 10 μL for 30 min at 37 °C at 300 rpm. Pre-amplification was performed with KAPA HiFi HotStart PCR Kit (2469088, UTECH) and Nextera PCR Primers (5 cycles). Quantitative PCR amplification was done with NEBNext Ultra II Q5 Master Mix (FC-121-1030, Alliance Global), Nextera PCR Primer and SYBR Gold to determine the number of additional cycles following which PCR amplification of additional cycles was performed. PCR fragments were purified with Ampure XP Magnetic Beads (A63880, Beckman Coulter, Agencourt, Indianapolis, IN, USA) to remove contaminating primer dimers, and library concentration and quality were determined by Agilent High Sensitive DNA Kit (5067–4626, Agilent, Santa Clara, CA, USA) and Bioanalyzer, respectively. Libraries were sequenced on Nextseq 550 to generate 12–21 million 150 bp paired-end reads per sample (Appendix A). Reads were aligned to the reference genome Danio rerio GRCz10 with BWA-MEM, then alignment files were sorted, and mitochondrial reads removed. Alignment rates were between 96.45% to 98.65% (Appendix A). ATAC-Seq QC, including enrichment plot and fragments summary, was performed with deeptools [65] and ATAC-Seq QC [66]. Peaks were called with macs2 by setting BAMPE mode, q-value as 0.05, and keep-dup all. To visualize fragments signal, the bam file was converted to a bw file with –smoothLength as 300 bp and –extendReads as 200 bp. After calling peaks, 3 samples showed robust consistency (Appendix A). We combined reads from 3 replicates together to increase Signal/Noise ratio increase peak number. The datasets are available in GEO (GSE173792).

### 2.6. Reduced-Representation Bisulfite Sequencing

RRBS was performed on ~80 ng of genomic DNA extracted from 40 WT livers at 120 hpf from 3 different clutches. MspI (100′000 U/mL, New England Biolabs, Ipswich, MA, USA) digested gDNA was used for preparing library, as previously described [67], with some modifications. To avoid loss of gDNA, after MspI digestion, end repair, and A-tailing, the reactions were stopped by heat inactivation. The adaptors used for multiplexing were purchased separately (Next Multiplex Methylated Adaptors-New England Biolabs). Libraries were size-selected by dual-step purification with Ampure XP Magnetic Beads (5067–4626, Beckman Coulter, Agencourt) to specifically select a region of fragments from 175 bp to 670 bp. Bisulfite conversion was performed with Lightning Methylation Kit (D5030, ZYMO Research) by following the manufacturer’s instructions. Libraries were amplified using KAPA HiFi HotStart Uracil+ Taq polymerase (KK2801, Roche, Basel, Switzerland) and purified with Ampure XP Magnetic Beads (5067–4626, Beckman Coulter, Agencourt, Indianapolis, IN, USA) before sequencing. Libraries were sequenced using the Illumina Nextseq 550. Fastq files are available in GEO (GSE173792). Quality control of the RRBS sequencing data was assessed using FASTQC and Trimmomatic and aligned to the reference genome GRCz10 as described previously [33].

### 2.7. Bioinformatic Analysis

RNA-Seq data was used to compare genes expressed at 78 hpf to 120 hpf based on statistical results, as mentioned in the RNA-Seq section from biological replicates (2 and 3, respectively; GSE130800 and GSE156420) and to examine the expression of genes in *uhrf1* mutants at 120 hpf (GSE160710). Zebrafish gene names were converted into human gene names by using Biomart for gene ontology (GO) analysis and Ingenuity Pathways Analysis (IPA). GO enrichment analysis was conducted using the GO hypergeometric over-representation test in the ‘ClusterProfiler’ package in R. REVIGO [68] was subsequently used to eliminate redundant enriched terms for different clusters of gene expression. An adjusted *p*-value < 0.05 was considered significant for all analyses.

To compare ATAC-seq with RNA-seq, genomic regions occupied by ATAC-seq peaks and not covered by ATAC-seq peaks were identified and overlapped with gene promoters (identified by using ‘genomation’ package in R and a window of +/−1000 bp around the TSS), defining “genes inside ATAC” and “genes outside ATAC”. For each category of genes, a violin plot of read counts was plotted.

Called peaks from ATAC-Seq reads were annotated based on a genomic element by using the annotatePeak function in ChIPseeker [69] based on the danRer10 reference in UCSC, implemented in Bioconductor package. Annotated peaks were categorized as covering promoter, intron, exon, or intergenic. Genes were categorized as ATAC-seq positive if a called peak overlapped with the gene body, the promoter, or for peaks that are designated as intergenic, the closest TSS to the ATAC-Seq peak was included. These ATAC-seq positive genes were analyzed for their GO category using Bioconductor as described for gene expression analysis and IPA (www.ingenuity.com, accessed on 29 June 2021). The top 20 GO categories designated as most statistically significant are represented.

RRBS data was analyzed for CpG methylation levels using the R package ‘methylKit’ [70]. CpGs covered at least 10 times in one biological replicate were included in the analysis. CpGs with methylation levels below 20% were treated as unmethylated, and above 80% were considered methylated. Genomic element annotation of CpGs was performed with R package ‘genomation’. To compare ATAC-seq with RRBS, genomic regions covered by ATAC peaks were selected, and DNA methylation levels were plotted for each genomic element. For plotting and statistical analysis, R package ‘ggplot’ and GraphPad Prism software were used. The code used for analysis is publicly available on Github, accessed on 20 June 2021 (https://github.com/zcmit/NYUAD_Sadler-Lab/tree/master/Nuclear%20organization%20in%20the%20developing%20zebrafish%20liver%20requires%20Uhrf1).

### 2.8. Immunofluorescence

120 hpf control and *uhrf1^−/−^* zebrafish embryos containing the *Tg(fabp10a: nls-mcherry)* transgene to identify hepatocyte nuclei were fixed in 4% paraformaldehyde for 4 h at room temperature, washed in PBS, and treated with 150 mM Tris-HCl at pH 9.0 for 5 min, followed by heating at 70 °C for 15 min according to an established protocol [71]. The embryos were then cooled, washed in PBS, the livers dissected out of the larvae, and permeabilized with 10 µg/mL Proteinase K (740506, Macherey-Nagel, Düren, Germany) in PBS containing 0.1% tween (PBST) for 10 min. After livers were washed 3 times with PBS, they were incubated in a blocking solution made up of 5% Fetal Bovine Serum (26140079, GIBCO) in PBS for 60 min at room temperature. The blocking solution was removed, and the livers were then incubated in 100 μL LaminB2 (ab8983, Abcam, Cambridge, UK) antibody in blocking solution (1:200 dilution) overnight at 4 °C. After 3 washes in PBST, the livers were incubated in secondary antibody (A-21236, Molecular Probes, Eugene, OR, USA) in blocking solution (1:400 dilution) in the dark for 2 h on a shaker. After 5 serial washes with PBST, the nuclei were counterstained with Hoechst (H3569, Thermo Fisher Scientific) diluted 1:1000 in PBS, washed in PBS 2 times to remove excess Hoechst, and mounted on a microscope slide with Vectashield (H1000, Vector Laboratories, Burlingame, CA, USA) and covered with a 0.1 mM coverslip for imaging using Leica SP8 confocal microscope. LAS X software (Leica software) was used for quantification from 3 separate optical sections per liver, which were then averaged from 3 livers per clutch per condition, and 3 clutches per sample were analyzed. Results were plotted in GraphPad Prism 8.

### 2.9. Confocal Imaging, Image Processing, and Analysis

Control and *uhrf1* mutant zebrafish embryos with the *Tg(fabp10a: nls-mcherry)* transgene that marks the hepatocytes were collected at 78 hpf, 96 hpf, and 120 hpf, fixed in 4% paraformaldehyde (15710, EM grade, Electron Microscopy Sciences, Hatfield, PA, USA) for 3 h at room temperature, and washed several times in PBS. Livers were microdissected, stained with Hoechst (1:3000 dilution in PBS, 33258, Sigma-Aldrich, St. Louis, MI, USA) for 10 min followed by 3 washes in PBS, and mounted on glass slides with Vectashield (H1000, Vector Laboratories, Burlingame, CA, USA) and coverslip.

Confocal imaging was performed using Leica TCS SP8 microscope with a 40× or 63× oil immersion at a scan speed of 100 Hz. Z-stacks were acquired using the galvo stage, with 2 µm intervals. Bit depth was 12, and to enhance image quality, field of view and laser intensity were adjusted separately for each sample. The acquired images were visualized during experiments using LASX software (Leica Application Suite X Leica microsystems CMS GmbH, Wetzlar, Germany). 3D analysis of Z-stacks Hoechst-stained nuclei was performed using the interactive 3D measurement tool in LASX for volume, surface area, sphericity, elliptical mean, and nuclei counting. Z-stacks were compiled into a 3D image, were adjusted for threshold and noise to define the nuclei clearly, set for minimum size to be measured as 1000 voxels, and measurements of the previously mentioned parameters obtained and exported in excel spreadsheets. For each time point and parameter measured, around 250 nuclei were sampled from at least 3 livers per condition from 3 individual clutches (at least 9 embryos).

### 2.10. Statistical Analysis

All experiments were carried out on multiple larvae from at least 3 clutches, as indicated in each figure and figure legend. Statistical significance included a two tailed Students *t*-test with adjustment for multiple comparisons as required. All plots along with statistical analysis were generated in GraphPad Prism 8. Bioinformatic analysis and visualization of genomic data were performed and plotted in RStudio.

## 3. Results

### 3.1. Hepatocyte Nuclear Morphology Evolves during Hepatic Outgrowth

We asked if hepatocyte nuclear morphology changed during hepatic outgrowth by dissecting livers from zebrafish larvae at 80 hpf, 96 hpf, and 120 hpf, staining with Hoechst, and visualizing using confocal imaging (Figure 1A). Z-stacks were used for 3D reconstruction to investigate nuclear shape and morphology (Figure 1B). In mammalian livers, albumin positive cells with large round nuclei were deemed to be hepatocytes [72,73]. In zebrafish larvae, cells were clearly identified as hepatocytes based on round morphology and larger size compared to non-hepatocytes. The use of the *Tg(fabp10a: nls-mCherry)* line in which the hepatocyte nucleus is labeled confirmed this designation (Appendix A). Z-stack reconstruction and deconvolution was used to examine nuclear shape. This showed that the round nuclei at 80 hpf increased in sized, and transformed to an elliptical shape as development proceeded (Figure 1B).

To quantify how nuclear size and shape changed during developmental time, 3D reconstructions for hundreds of nuclei in livers from multiple clutches at each stage were used. Hepatocyte nuclei had an average volume of ~120 μm^3^ and surface area of ~200 μm^2^ at 80 hpf, which increased to a volume ~220 μm^3^ and surface area of ~400 μm^2^ by 120 hpf (Figure 1C,D). The nuclear shape as measured by sphericity and ellipsoid mean changed as the hepatocytes matured, with mostly spherical nuclei observed at 80 hpf with ~0.5 sphericity and ~3 μm ellipsoid means transforming to an elliptical shape by 120 hpf with ~0.4 sphericity and ~4 μm ellipsoid mean (Figure 1E,F). These data uncover the distinct morphological alterations (elongation) that hepatocyte nuclei undergo during development, reflecting the transition from a proliferative early-stage hepatocyte to a mature hepatocyte.

We next assessed landmarks of nuclear morphology, including nuclear lamins, which maintain nuclear structure and serve to package silenced genes in inaccessible heterochromatin [74], the nucleoli where ribosome biogenesis occurs, and prominent heterochromatin foci. Light and diffuse Hoechst staining represent euchromatin, and intense Hoechst foci represent heterochromatin. Heterochromatic DNA at the hepatocyte nuclear lamina is observed as a prominent ring in the inner membrane of the nuclei, and nucleoli were identified as large dark spots surrounded by a thin ring of heterochromatin within the nucleus (Figure 1A). We quantified these observations in hundreds of nuclei from multiple larvae from at least two clutches collected on each day during hepatic outgrowth. The number of nucleoli varied between 1 to 4 during early liver development (80 hpf and 96 hpf), but by 120 hpf, a majority of hepatocyte nuclei had one nucleolus (Figure 1G), and a prominent lamina was present in most hepatocyte nuclei at all stages (Figure 1H). Hence, hepatocyte maturation is characterized by nuclear enlargement, elongation, and reduction in nucleoli number. This shows that even after acquiring many features of differentiated cells, the structure and size of the hepatocyte nucleus continues to change during hepatic outgrowth.

### 3.2. Open Chromatin in the Larval Liver Is Enriched for Developmental Genes

To define the chromatin landscape in the liver of 120 hpf zebrafish larva, we optimized an established protocol for generating ATAC-Seq libraries [64] to be useful for this tissue. The most reproducible protocol used livers dissected from 10 *Tg(fabp10a:CAAX-EGFP)* larvae at 120 hpf, where GFP expression was used to verify that the majority of cells were hepatocytes; however, a minority population of cells in the isolates did not express GFP and were likely the other cell types that populate the 5 dpf liver. Single cells were obtained by dissociation with 2.5% Trypsin, which yielded on average 10,000 cells, with 95% viability based on trypan blue exclusion (Appendix A). These cells were lysed and used immediately for ATAC-Seq library preparation following the established protocol [64]. Libraries were inspected for fragment length distribution, and those with the expected nucleosomal laddering pattern enriched around 200, 350, 550 bp, respectively (Appendix A) were selected for sequencing. Through optimization, we found that libraries prepared with freshly isolated nuclei from 10,000 cells or more, typically yielded high quality libraries, but as few as 500 cells can yield usable libraries (Appendix A).

We obtained between 12–21 million reads per each of three ATAC-Seq libraries made from biological replicates, with over 96% alignment rate for all reads (Appendix A). Enrichment quality control showed read density peaks close to 200 bp in all three samples (Appendix A, Appendix A), which indicated that mono-nucleosomes are most captured. To increase the signal to noise ratio and the coverage of open chromatin, qualified aligned reads in all three replicates were combined to call peaks [75]. This generated 9758 called peaks that covered 4,425,002 bp, representing 0.32% of the zebrafish genome. In the human genome, the accessible genomic region comprises ~2–3% of the total DNA sequence yet captures more than 90% of regions bound by TFs [76], and this is comparable to the accessible genome in many zebrafish tissues [51]. We speculate that the landscape of chromatin accessibility in zebrafish liver defined by this ATAC-Seq should also encompass regulatory regions and highly expressed genes and exclude regions of heterochromatin, and future analysis will focus on comparison to other zebrafish tissues with more refined chromatin maps [51].

We investigated the distribution of ATAC-Seq peaks across genomic elements using the reference genome GRCz10 annotation for introns, exons, and intergenic regions and added proximal promoters, which we defined as +/−1000 bp from the TSS of each annotated gene (Figure 2A). We found 3% of the detected accessible chromatin covered promoters, while intergenic and introns accounting for 76% and 20%, respectively, representing an enrichment for promoters, as these are expected to encompass less than 2% of the genome (Figure 2A). Introns and the intergenome are highly populated by repetitive sequences, including transposable elements and simple repeats. We were surprised that a majority of ATAC-Seq peaks fell within these genomic elements, and we, therefore, examined the distribution of (TEs) elements in the open and closed chromatin, defined as regions not covered by ATAC-Seq peaks. A pie chart showing ATAC-Seq peak distribution over different TE classes compared to non-TE sequences (Figure 2B) shows that over 40% of detected open chromatin covers TEs. This suggests that the TEs that fall in these open regions have mechanisms other than packaging in heterochromatin as a mechanism for silencing them. An example is shown in Figure 2C, where a DNA transposon is encompassed by an ATAC-Seq peak covering the *fabp10a* promotor, yet is not expressed. This is in contrast to the fatty acid binding protein 10a (*fabp10a*) gene, which is highly expressed in hepatocytes [77,78] and is in open chromatin as expected. This confirms that the quality of the ATAC-Seq libraries was sufficient to identify major chromatin accessibility patterns, even at low sequencing depth (Appendix A).

We asked whether genes that carry out distinct functions were clustered into open chromatin using GO analysis of molecular functions. Genes functioning in nucleic acid binding, signaling, and metabolism were highly enriched (Figure 2D). Interestingly, Wnt signaling, which is essential for liver development [79], was enriched (Figure 2D). Ingenuity Pathway Analysis of these same genes also highlighted genes in the Wnt signaling pathway and other signaling pathways involved in developmental processes in addition to genes and pathways involved in pluripotency and cancer. These could indicate that hepatocytes at 120 hpf maintain potency to change fate and to receive instructive signals that guide developmental processes (Figure 2E). In summary, open chromatin in zebrafish livers encompasses gene regulatory regions of developmental genes, and some TEs also reside in open chromatin. This suggests that at 120 hpf, the chromatin landscape in the zebrafish liver remains competent for receiving developmental signals.

### 3.3. Hepatic DNA Methylome Is Enriched in the Intergenome and on Transposons

DNA methylation is one of the best-characterized epigenetic marks and is associated with heterochromatin, the integrity of the nuclear lamina, and nucleolar structure [7,80]. An intriguing possibility is that establishing a cell type-specific DNA methylation pattern could be a scaffold for establishing the nuclear structure that characterizes each cell type. To define the zebrafish larval liver DNA methylome, we performed Reduced Representative Bisulfite Sequencing (RRBS) on genomic DNA of three biological replicates of pooled 120 hpf WT livers. We combined the three replicates to maximize the amount of CpGs covered, and we detected 2,173,502 CpGs, which covers 9.08% of the total CpGs in the zebrafish genome. As expected, methylated CpGs were enriched at intergenic regions compared to the proportion of CpGs present in the whole genome (Figure 3A). Promoters (+/− 1000 bp surrounding TSS) were enriched for unmethylated CpGs (<20%), while methylated CpGs (>80%) were mainly found in intergenic regions and introns (Figure 3A). CpG methylation showed a bimodal distribution, with nearly all CpGs categorized as fully methylated or not methylated (Figure 3B). This is consistent with the observation that in terminally differentiated tissues, CpGs are static as either methylated or not methylated in all cells, and that domains of partially methylated CpGs are exception and hallmark of disease state, such as cancer [81,82]. Consistent with a critical role for DNA methylation in suppressing TEs, CpGs captured by RRBS were enriched in LINEs and LTRs (Figure 3C). Comparing the level of methylation inside transposable (DNA, LTR, LINE, SINE) and non-transposable (non-TE) elements showed that these TE classes were highly methylated compared to non-TE elements (Figure 3D). We previously showed that TE expression in the zebrafish liver is repressed, in part, by DNA methylation [33], and this supports a simple model whereby TEs are repressed by packaging into constitutive heterochromatin [83].

DNA methylation is predicted to occupy closed chromatin, which was confirmed in the liver by integrating RRBS with ATAC-Seq. This showed that the CpGs inside ATAC-Seq peaks (11,034 CpGs) were less methylated compared to a random selection of the same number of CpGs outside peaks (Figure 3E). We next investigated the correlation between ATAC-Seq, DNA methylation levels, and genomic elements by dividing the CpGs inside or outside ATAC-seq peaks based on their annotation (i.e., promoters, exons, introns; Figure 3F). As expected, we found that promoters were not methylated in either the open or closed chromatin, but that exons in open chromatin regions were significantly less methylated than exons outside of ATAC-Seq peaks. Introns in both open and closed chromatin regions have a high level of DNA methylation. These data indicate that DNA methylation in zebrafish liver decorates the TEs in intergenic regions and gene bodies as previously described also for other tissues in many vertebrate animals [84]. As found in other tissues, intergenic and intronic regions, largely consisting of TEs, showed the highest level of DNA methylation detected.

### 3.4. Transcriptome of the Late Developing Liver Reflects Hepatic Maturation

Nuclear structure and the chromatin landscape dictate gene expression. We analyzed gene expression changes during hepatic outgrowth using RNA-Seq analysis on pools of livers collected at 78 and 120 hpf, capturing 25,455 genes expressed at least in one sample with a Read Count > 0 (Appendix A). There were 1837 genes expressed at higher levels at 120 hpf (log2FC > 1.5 and padj < 0.05) and 2017 genes expressed at lower levels (log2FC < −1.5 and padj < 0.05), while a majority of genes (14750) were unchanged between 78 hpf and 120 hpf (Figure 4A,B). To determine whether the 120 hpf liver achieved a gene expression profile compared to a fully mature liver, we compared the transcriptome from this time point to a published dataset from adult zebrafish liver [51]. Of all the genes expressed, each time point showed 53% common to both datasets, with nearly all the genes expressed in the 120 hpf liver also expressed in adults (Appendix A). It is possible that the difference in genes captured in the adults compared to the 120 hpf samples reflects the increased representation of other hepatic cell types in adult samples, such as more extensive biliary cell and endothelial cell branching or the maturation of stellate cells [85] or it could reflect deeper sequencing depth in the adult samples captured more genes.

To identify the specific pathways that were changed at 120 hpf compared to 78 hpf, GO analysis was used to determine the functions of differentially expressed genes (Figure 4C). The 1837 genes upregulated at 120 hpf were enriched for genes involved in liver function, metabolism, and transport processes, while the 2017 genes downregulated at 120 hpf were enriched for those involved in DNA replication, chromosomal and extracellular matrix organization, and molecular transport. The 14,750 genes that were unchanged were mainly involved in ribosome biogenesis and metabolic processes that serve cellular housekeeping functions and reflect central hepatocyte functions (Figure 4C and Appendix A). Genes for ribosome biogenesis were shared among the clusters, indicating the essential nature of these pathways at all time points analyzed and the high protein synthesis demand of differentiated hepatocytes. IPA analysis showed that genes expressed at higher levels in 120 hpf samples were enriched for signaling pathways important for hepatocyte function, such as FXR and LXR signaling, cellular growth, energy metabolism, and those that were higher at 78 hpf (down at 120 hpf), were DNA replication and maintenance of cell division (Figure 4D).

We integrated the 120 hpf RNA-Seq and 120 hpf ATAC-Seq datasets by selecting those genes with ATAC-Seq coverage of +/− 1000 bp around TSS (1143 genes). Of these, 418 were expressed at 120 hpf, and comparing the expression levels to an equal number of randomly selected genes whose promoters were not covered by ATAC-Seq (log10(read counts)) showed that genes residing in open regions had on average higher expression than those that did not (Figure 4E). These data indicate that, during outgrowth, genes involved in both energy metabolism and catabolism are maintained, while genes essential for cell division are turned off. As the liver matures, the transcriptome changes to reflect the metabolic activity essential for hepatic function. Moreover, GO analysis of genes that remain in open chromatin in the mature liver (120 hpf), but remain silenced are involved in the synthesis of bile acids, NR1H2, H3-mediated signaling pathways, and other pathways involved in biliary function (Appendix A). This could reflect the fact that at 120 hpf, hepatocytes remain capable of trans differentiation into biliary cells [86]. This could also reflect the maintenance of genes that require rapid activation upon injury in active chromatin states, as we recently showed is the case in the mature mouse liver [87].

### 3.5. Uhrf1 Loss Leads to Large Dysmorphic Hepatocyte Nuclei

The shape of a nucleus is governed mainly by the structure, and, in turn, epigenetic status of the chromatin and nuclear envelope [88]; however, it is not known how epigenetic patterning during development establishes cell-specific nuclear structural patterns. Zebrafish *uhrf1* mutants have global DNA hypomethylation, multiple cellular defects, including a small liver characterized by cell cycle arrest and apoptosis [31,52]. DNA methylation may, thus, be involved in establishing hepatocyte nuclear morphology during development. To test this, we assessed hepatocyte nuclear morphology in *uhrf1* mutants and their phenotypically WT siblings at 80 hpf, 96 hpf, and 120 hpf using confocal imaging (Figure 5A) and 3D reconstruction of Z-stacks (Figure 5B). As observed in WT larvae (Figure 1), hepatocyte nuclei in sibling controls were round at 80 hpf, progressing to a more elongated ovoid nucleus by 120 hpf (Figure 5A top panel and B). The *uhrf1* mutant hepatocytes were distinguished as larger and misshapen at every stage (Figure 5A,B). Quantification of these changes over hundreds of hepatocyte nuclei from *uhrf1* mutants and their phenotypically normal siblings showed significantly bigger nuclei in *uhrf1* mutants at all stages, both by volume (Figure 5C) and by surface area (Figure 5D), achieving over 1.5× the size of control nuclei at 120 hpf. *uhrf1* mutant nuclei were highly dysmorphic, with a scalloped surface indicated by the loss of sphericity (Figure 5B,E, 96 and 120 h) and an elongated shape (Figure 5F). We noted that the nuclear morphology changes do not seem to affect hepatic differentiation, even though the liver is significantly smaller, since RNA-Seq data from 120 hpf *uhrf1* mutant livers [33] showed that over 40% of the genes that were expressed in both 120 hpf and adult livers also were expressed in *uhrf1* mutant livers (Appendix A). GO analysis showed that these overlapping genes are involved in translation and metabolism (Appendix A).

The marked differences in *uhrf1* mutant hepatocyte nuclear structure were apparent as early as 96 hpf. *uhrf1* mutant nuclei, like their sibling controls, had multiple nucleoli at 80 hpf, and the reduction in nucleoli observed during hepatic outgrowth maturation in controls is not observed in *uhrf1* mutants, so that over 40% of all mutant nuclei have multiple, fragmented nucleoli at 120 hpf (Figure 6A). We analyzed RNA-Seq of 120 hpf *uhrf1* mutant livers for changes in nucleoli-associated genes. While most nucleoli specific genes associated with ribosome, RNA processing, and metabolic genes (*rpl3*, *rpl23a*, *akr1c4*, *akr1c2*, *akr1c3*, *dennd5b*, *pin4*) are downregulated, few genes (*nusap1*, *phf6*, *rpf1*) are upregulated (Figure 6B). We reconfirmed some of the nucleolar gene expression changes observed in RNA-Seq with real-time PCR analysis (Figure 6B and Appendix A). Except for *rpl23a* expression (Up in the *uhrf1^−/−^* livers by qPCR, but downregulated in RNA-Seq), expression of all other genes examined had similar trend as the RNA Seq. This is consistent with other findings that DNA methylation loss causes nucleolar disruption with impaired function [89,90].

Similarly, *uhrf1* mutant hepatocytes showed a well-defined nuclear lamina at 80 hpf, which progressively deteriorated to a fuzzy and indistinct ring around the nuclei as the embryo aged (Figure 5A). A well-defined Hoechst positive ring was detected in less than 40% of hepatocytes at 96 and 120 hpf (Figure 6C). Cytoskeletal protein Lamin B2 localizes primarily in the nuclear lamina across cell types and is required to maintain nuclear architecture [91,92,93]. Immunofluorescence for laminB2 of 120 dpf control livers revealed that nearly all hepatocytes have a well-defined, uniform laminB2 positive rim around hepatocyte nuclei (Figure 6D,E, top panel), whereas the staining was almost entirely lacking in *uhrf1* mutant hepatocytes. Similarly, *dnmt1* mutant hepatocytes at 120 hpf display a diffused, mild staining to no laminB2 staining at the nuclear lamina (Figure 6D,E bottom panel and F). These data indicate that *uhrf1* is required to reduce nucleoli, normal lamin formation, and genome organization in hepatocytes during hepatic outgrowth. Since the laminB2 staining is phenocopied by *dnmt1* mutants, we conclude that loss of lamin formation is attributed to a loss of DNA methylation.

## 4. Discussion

Organ development involves dramatic reorganization at the tissue and cellular level. These tissue-level changes are accompanied by reorganization of intracellular structures to meet the demand on cellular function acquired during fate change. Well-studied examples include the expansion of the endoplasmic reticulum and Golgi apparatus to accommodate the expanded secretory cargo as cells with high secretory protein output undergo differentiation, and the number and size of the nucleolus expand in cells with increased demand for ribosome biogenesis. In this study, we examined the changes in nuclear structure that occur during hepatic outgrowth and morphogenesis. We found that hepatocyte nuclear morphology is established soon after hepatocytes differentiate; however, significant reshaping occurs during hepatic outgrowth, with nuclei becoming more elliptical and larger. This is accompanied by a reduction in nucleoli number (nucleolar hypotrophy), coincident with decreased ribosome biogenesis and the cessation of cell proliferation [94,95]. By integrating ATAC-Seq and RRBS profiling, we demonstrate that DNA methylation and chromatin accessibility with respect to exons are inversely correlated in the zebrafish larval liver at 120 hpf, as expected for differentiated cells. RNA-Seq analysis from two time points during hepatic outgrowth show that as early as 78 hpf, most of the genes involved in central hepatocyte functions are expressed; as the liver matures, the gene expression profile reflects the further refinement of energy and small molecule metabolism and the loss of genes involved in cell growth, proliferation, and signaling, however comparison to adult liver transcriptome profile suggests that further hepatic maturation occurs as the fish age. Interestingly, genes related to cell signaling and response to injury remain in open chromatin configurations, which is similar to a pattern we recently discovered in the mouse liver [87]. We establish here that the size, shape, and nucleolar formation requires *uhrf1*, a key component of the DNA methylation machinery, and attribute this to loss of DNA methylation as the laminB2 loss in hepatocytes is phenocopied by *dnmt1* mutation. Our results suggest that Uhrf1 is important for genome organization in the nucleus and for establishing the structural components that contribute to the nuclear shape and cell type-specific morphology during hepatic outgrowth.

As expected, we found that ATAC-Seq profiling of the zebrafish larval liver was inversely correlated with DNA methylation and that genes that are important for hepatocyte formation (i.e., *wnt* signaling) and function (i.e., metabolism) reside in open chromatin. Interestingly, other genes that are not immediately recognized as key hepatocyte function genes are also covered by ATAC-Seq. Moreover, more than 2/3 of genes that are in close proximity to an ATAC-Seq peak in the liver are not expressed. This could reflect the retention of pluripotency of cells in the 120 hpf liver [96,97] or might also be impacted by the sequencing depth as the absence of an ATAC-Seq peak might be changed with deeper sequencing. Liver development is evolutionarily and genetically conserved across vertebrates [48]. A comprehensive study of chromatin dynamics during early mouse liver development showed that the landscape of histone modifications and chromatin accessibility change such that early development is dominated by an open chromatin and euchromatic histone marks, and as development progresses, the chromatin is less open and marked by the heterochromatic mark H3K9me3 [98]. In another study in a partial hepatectomy model of mouse, adult hepatocytes rapidly transition into a fetal-like liver chromatin landscape detected by ATAC-Seq, leading to the opening up of chromatin and expression of genes critical for fetal liver development [99]. Our recent finding that genes required for liver regeneration are silenced in quiescent livers, but maintained in an open chromatin state [87] suggests that the liver has a mechanism to package genes that are needed to respond to injury or other stimuli in a “ready-set-go” configuration so that they can be activated quickly when needed. This is supported by other studies that suggest that the adult liver utilizes epigenetic mechanisms to reacquire cells with progenitor-like properties to regenerate hepatocytes [99,100,101]. Similarly, we find that in the chromatin landscape of 120 hpf zebrafish liver, some of the open ATAC regions whose genes are silent at 120 hpf correspond to processes involved in biliary functions (not shown), suggesting the chromatin is part of the mechanism that maintains hepatocyte ability to transdifferentiate if required.

The close relationship between DNA methylation and a closed local chromatin structure is well established. While most of the methylome remains static across cell types, due to the heavy methylation of TEs in all cells, small but distinct differences in DNA methylation patterns associate with different cellular identities [102,103]. Our DNA methylation data from larval zebrafish livers show, unsurprisingly, that most gene promoters remain unmethylated as in nearly all other vertebrate tissues examined [104], and that methylated CpGs are enriched in TEs and in gene bodies, as consistently reported in the genome of vertebrates [105,106]. The zebrafish genome has a remarkably rich TE content, with approximately 65% of the TEs expressed during development [107]. DNA methylation is, thus, a conserved mechanism in repressing the transcription of especially evolutionarily young TEs [108]. Therefore, we conclude that DNA methylation impacts the nuclear organization, but that the effects on gene expression are indirect through either an altered nuclear organization or, as we showed previously, reflect TE activation or the cellular defects that occur as a result of *uhrf1* mutation [29,33].

*uhrf1* loss in the liver causes other cellular responses: DNA damage, continuous DNA replication that fail to progress through the cell cycle and cell death, preventing organ expansion in these embryos [31,52]. In differentiated cells, the nuclear shape is mediated through heterochromatin-cytoskeletal contacts in the nuclear lamina [74,109,110]. In previous studies, we discovered robust activation of retrotransposons in *uhrf1* mutant zebrafish livers, which we attributed to the loss of DNA methylation in this model [29]. While this latter result suggests that unleashing transposons in this mutant could contribute to the cellular defects that cause hepatic outgrowth failure, these could also be related to changes in heterochromatin structure. Recent findings reveal overlap of H3K9me2 (Large Organized Chromatin Lysine Modifications, LOCKs), associated lamina domains (LADs), and 5MeC in the vertebrate genome [111,112], implicating heterochromatin marks, nuclear lamina, and DNA methylation as intimately connected. Our results show the importance of these features in the context of the final stage of liver development and indicate that Uhrf1 is required for these processes.

Heterochromatin regions are condensed, stiffer, and are, thus, deformed less than euchromatin [113], making chromatin dynamics an important determinant of nuclear shape. Tethering of chromatin with the nuclear lamina is important for conferring nuclear structure and stiffness [114], and chromatin confirmation changes contribute to the mechanical changes of the nuclear envelope [115]. In vertebrates, 5MeC, LOCKs, and LADs decorate the laminar rim and repress the lamin-associated genes [1,116,117,118,119]. The loss of heterochromatin at the nuclear rim and depletion of laminB2 in *uhrf1* and *dnmt1* mutants is a likely cause of the deformed nuclear shape, and we predict, result in reduced nuclear stiffness. Similar results linking chromatin state and nuclear morphology have been done mainly in cell lines [91,120,121], and our work establishes this link during hepatocyte development.

Studies suggest that the nuclear lamina have a vital role, not only in the maintenance of nuclear shape and structure [122,123,124], but also in transcriptional regulation, nuclear pore positioning and function, DNA, and heterochromatin organization [125,126,127,128,129,130]. Lamins are also implicated in DNA replication [131], and DNA damage response [132], and laminopathies are associated with DNA damage [133]. Interestingly, patients with Lamin A mutations show rearrangement of LADs that correlated with altered DNA methylation and marked changes in gene expression [134]. While *uhrf1* zebrafish mutants do not display the cardiac or muscular defects found in laminopathy patients, a zebrafish model of depleted lamin A shows some phenotypes found in *uhrf1* mutants, including more cells in S-phase, dysmorophic nuclei, and deregulated cell cycle genes [135]. This suggests that the cell cycle and DNA damage phenotypes observed in *uhrf1* deficient cells [31] could be related to lamin or LAD defects.

Our finding that *uhrf1* mutant hepatocytes maintain a high level of nucleoli during hepatocyte maturation is likely directly related to a loss of DNA methylation. This is supported by the finding of disorganized nucleoli, which become fragmented into small nuclear masses in tissue culture cells lacking Dnmt1 [89]. Thus, the finding of increased nucleoli numbers in the *uhrf1* mutant hepatocytes could be caused by nucleoli fragmentation, due to loss of DNA methylation as a key structural component of the nucleolar rim [136]. Another possibility is that during hepatic outgrowth, when hepatocytes are highly proliferative, nucleoli may be reformed so that the material from multiple nucleoli are combined to form a few big nucleoli. In human tumor tissue studies, nucleoli size, number, and shape were found to increase in cancer growth and malignancy, and this was attributed to not only to increased ribosomal biogenesis, but also with DNA replication as the nucleolus contains both RNA (12–15%), and DNA (2–10%) [137,138,139]. Since cell cycle progression is blocked in hepatocytes of *uhrf1* mutant zebrafish [31], it could be that this also contributes to the failure to reduce the nucleolar number in these mutants.

Global DNA hypomethylation is the first observed abnormality in cancer [140,141,142], indicating this as a potential cause of cancer. In many types of cancer, widespread reshuffling and loss of large stretches of condensed DNA marked by H3K9me2 and lamina-associated domains (LADs) have been reported [119,142,143,144]. Interestingly, regions of hypomethylated DNA found in cancer cells correspond to these same domains [112,142,143,145], indicating synergy between them. Cancer cells are also frequently identified by major changes in the structural landmarks in the nucleus, by genome wide loss of DNA methylation with regions of local hypermethylation, and in many cases, the marked differences in nuclear shape, size, and the prominence of structures, such as the nucleolus are pathognomonic for the type and stage of the cancer. UHRF1 is highly expressed in many types of cancers [146] and is an oncogene in the liver [37]. Moreover, as liver cancer is characterized by a global DNA hypomethylation [147] atypical irregular nuclear structure [148], changes in nuclear size [149], and progenitor-like features [150,151], it is possible that the high levels of UHRF1 in cancer cells function as a dominant negative, blocking DNA methylation and deposition of other heterochromatin marks, reverting the nuclear morphology and function to a progenitor-like state [152,153]. These studies combined with the current findings propose the intriguing possibility that high levels of UHRF1 in cancer could act as a dominant negative, recapitulating some of the cellular and gene expression features that are found in hepatocytes that lack *uhrf1*.

This study represents an integration of advanced imaging and genomics to describe how nuclear structure, genome organization, DNA methylation, chromatin accessibility, and gene expression occur during normal liver development. This both establishes the process of hepatocyte nuclear organization during hepatic outgrowth and implicates DNA methylation mediated by Uhrf1 as a driver of this process. This is relevant to understanding the mechanisms of development, as well as how cancer cell morphology impacts malignant potential.

## Figures and Tables

**Figure 1 genes-12-01081-f001:**
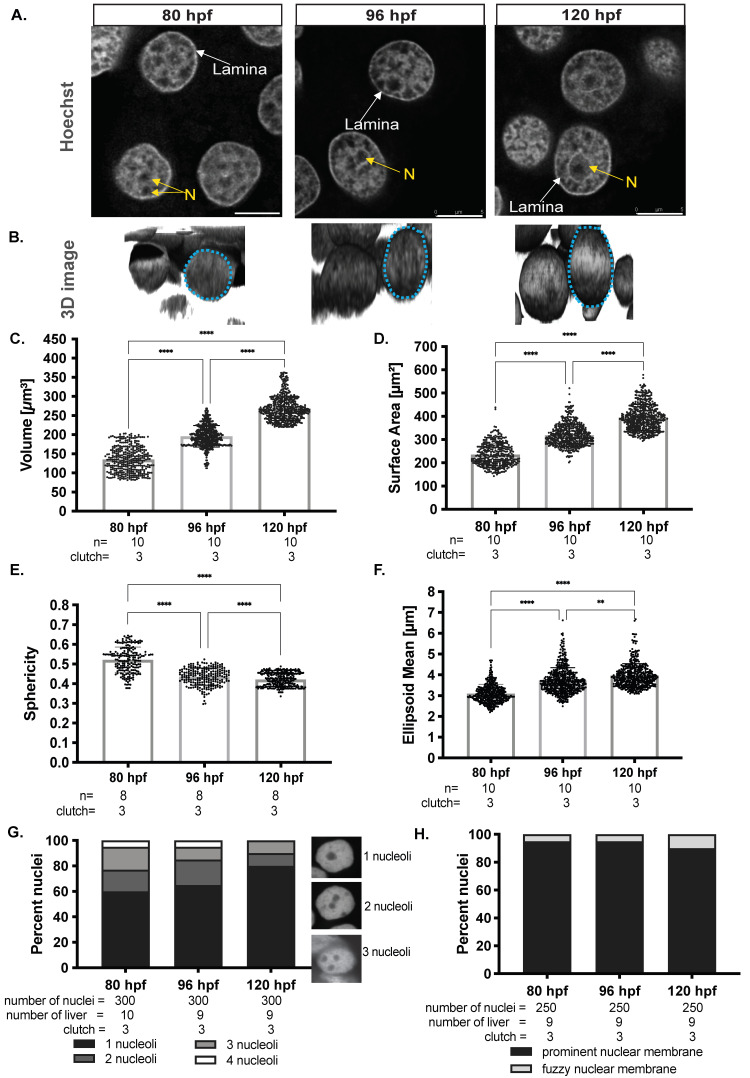
Hepatocyte nuclei increase in size, become elliptical and reduce nucleoli number during hepatic outgrowth. (**A**). Representative confocal images of Hoechst-stained hepatocytes from larvae at 80, 96, and 120 hpf. Regions depleted of Hoechst staining represent euchromatic regions, strong Hoechst foci represent heterochromatin. Yellow N indicates nucleolus; White arrows point to examples of the prominent nuclear lamina. Scale bar: 5 μm (**B**). 3D reconstructions of Z-stacks of Hoechst-stained hepatocyte nuclei at 80, 96, and 120 hpf display changes in size and shape as the embryo ages. Morphological measurements obtained from hepatocyte nuclei were analyzed using Las-X software for volume (**C**), surface area (**D**), sphericity (**E**), and ellipsoid mean (**F**). Each dot on the graph represents an individual nucleus, and the horizontal represents the median Hepatocyte nuclei were scored for the number of nucleoli per hepatocyte (**G**) and the presence of normal nuclear lamina observed as a rim around the nucleus by Hoechst staining (**H**). All measurements except sphericity were obtained by automated measurements in LasX software, while the sphericity was calculated based on surface area and volume measurements. All analyses were performed on larvae from 3 clutches, with minimum of 3 livers analyzed per clutch. ** *p* < 0.005, **** *p* < 0.00005. L–nuclear lamina, N-nucleoli. Magnification: 63×, 4× zoom, Scale bar: 5 μm.

**Figure 2 genes-12-01081-f002:**
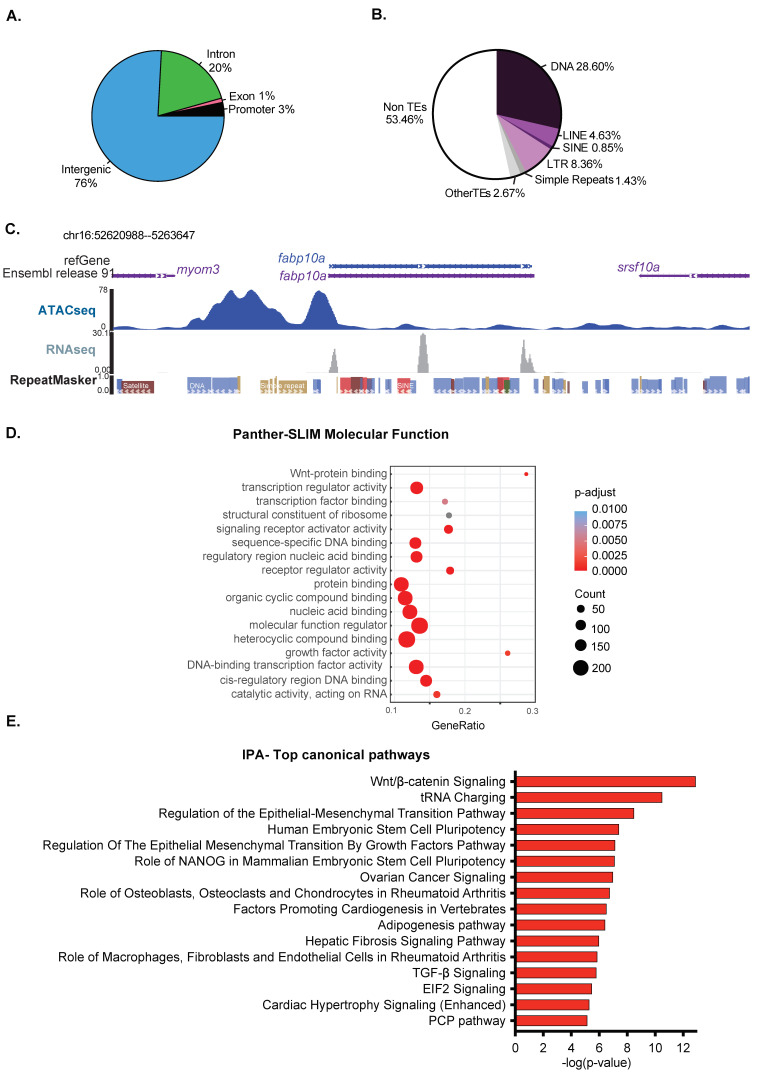
The chromatin landscape in 120 hpf zebrafish livers remains competent for development. (**A**). ATAC-Seq peaks in the 120 hpf zebrafish liver were categorized based on the proportion in distinct genomic elements designated as a promoter (1 kb +/− TSS), exon, intron, and intergenic. In total, 575 promoters are covered by ATAC-Seq peaks. (**B**). The relative proportion of transposable elements (TEs) across open chromatin region in zebrafish liver at 120 hpf. TE families, including DNA, LINE, SINE, LTR, and simple repeats, are represented with different colors. The other types of REs and non-RE regions are represented as others. (**C**). Browser view of ATAC-Seq and RNA-Seq from 120 hpf livers from the *fabp10a* gene locus and surrounding regions. Note the DNA transposon located in the center of the ATAC-Seq peak is not expressed in this sample. (**D**). Gene Ontology analysis of the molecular function of genes that either overlap with ATAC-Seq peaks or are the nearest to a peak falling in intergenic regions. The canonical process with annotated genes from open chromatin regions. The top 20 categories from each section were plotted. Dot size in (**D**) represents the number of overlap input genes with the specific geneset. (**E**). IPA analysis of the pathways that are enriched in the same ATAC-Seq positive genes used in (**D**).

**Figure 3 genes-12-01081-f003:**
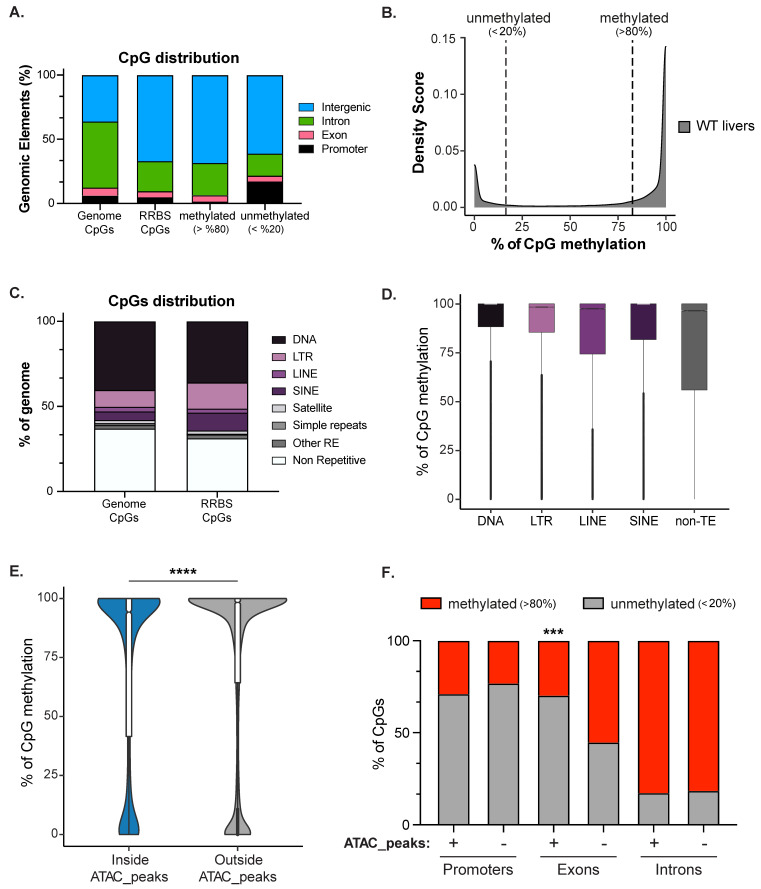
DNA methylation is enriched in the intergenome and on transposons. (**A**). RRBS Genomic Annotation on 120 hpf zebrafish livers of the CpGs common to the unified dataset of 3 biological replicates from 120 WT zebrafish larvae (2,173,502 CpGs). CpGs were categorized as methylated (>80%; 1,521,222 CpGs) and unmethylated (<20%; 427,492 of CpGs) and then were classified based on annotated genomic elements. The distribution of CpGs in the whole genome is plotted as a comparison. (**B**). Density plot showing the distribution of CpGs based on methylation level. (**C**). Annotation of CpGs contained in the Repetitive Elements (REs) or not-RE in zebrafish genome and RRBS dataset. (**D**). Box plot displaying the percent of methylation of CpGs contained in each Class of Transposable Elements (TEs) and not-TEs. (**E**). Violin plot displays the percent of CpG methylation in zebrafish livers contained inside (blue) or outside (grey) ATAC-Seq peaks. An equal number of CpGs (11,034 CpGs) contained outside of ATAC-seq peaks were randomly selected for comparison. **** indicates *p*-value < 0.0001 calculated by unpaired non-parametric Mann–Whitney test. (**F**). Box plot displaying the percent of CpGs either methylated (red, >80%) or not-methylated (grey, <20%) in WT livers: From left, CpGs contained inside (represented with +) or outside (represented with −) of ATAC-Seq peaks classified based on their location in annotated genomic element (promoters, exons, and introns). *p*-values were calculated by unpaired non-parametric multiple comparisons Kruskal-Wallis test. *** means *p*-value < 0.001.

**Figure 4 genes-12-01081-f004:**
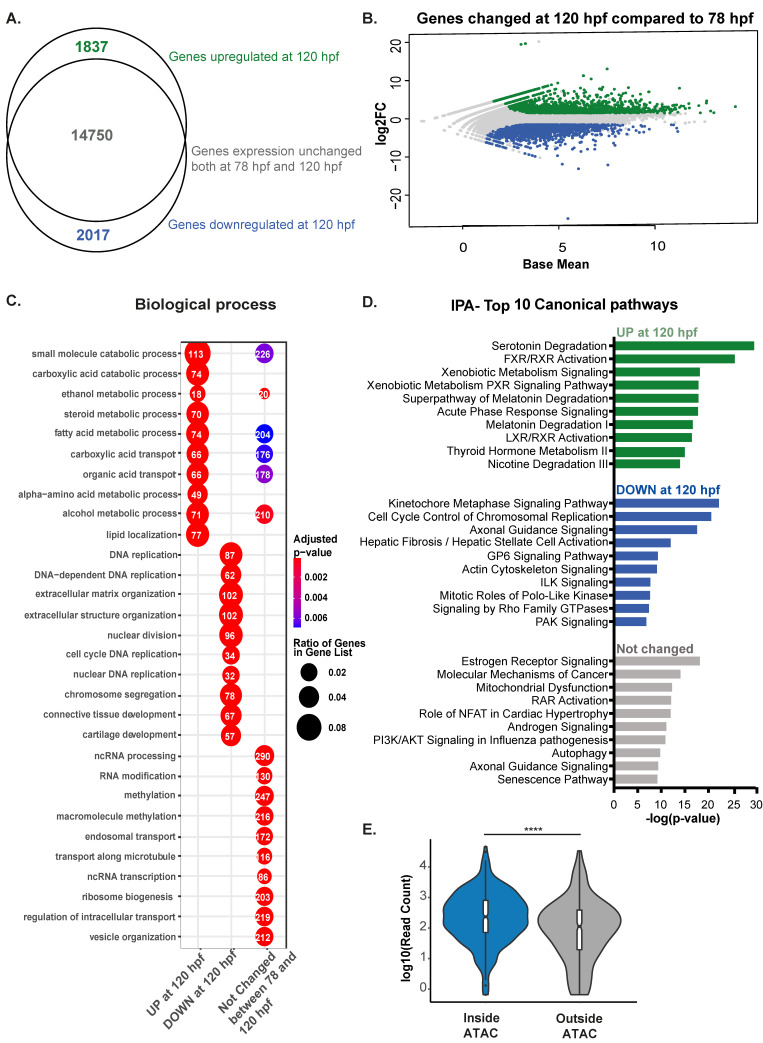
Expression of genes involved in signaling pathways enriched during early hepatic outgrowth. (**A**). RNA-Seq comparative analysis on differential expression of 120 hpf zebrafish livers compared to 78 hpf livers classified as expressed based on log2FC > 1.5 and padj < 0.05 represented by Venn diagram showing that most genes are already expressed at 78 hpf. (**B**). MA plot showing log2 fold change of genes at 120 hpf calculated on 78 hpf WT livers and Base Mean. In green upregulated genes (padj < 0.05 and log2FC > 1.5), in blue downregulate genes (padj < 0.05 and log2 Fold Change <−1.5) and in grey all other genes. (**C**). Gene Ontology of Biological Processes for genes categorized in B, with the 10 most significant terms for each group plotted. Numbers in each circle represent the number of genes of each GO term that are captured in the dataset plotted. (**D**). IPA of top 10 canonical pathways enriched for each group. (**E**). Violin plot displaying expression values of genes contained in open regions (inside ATAC) and in closed regions (outside ATAC) indicates that genes present in open regions are more expressed than genes not covered by ATAC-seq. **** indicates *p*-value < 0.0001 measured by unpaired *t*-test.

**Figure 5 genes-12-01081-f005:**
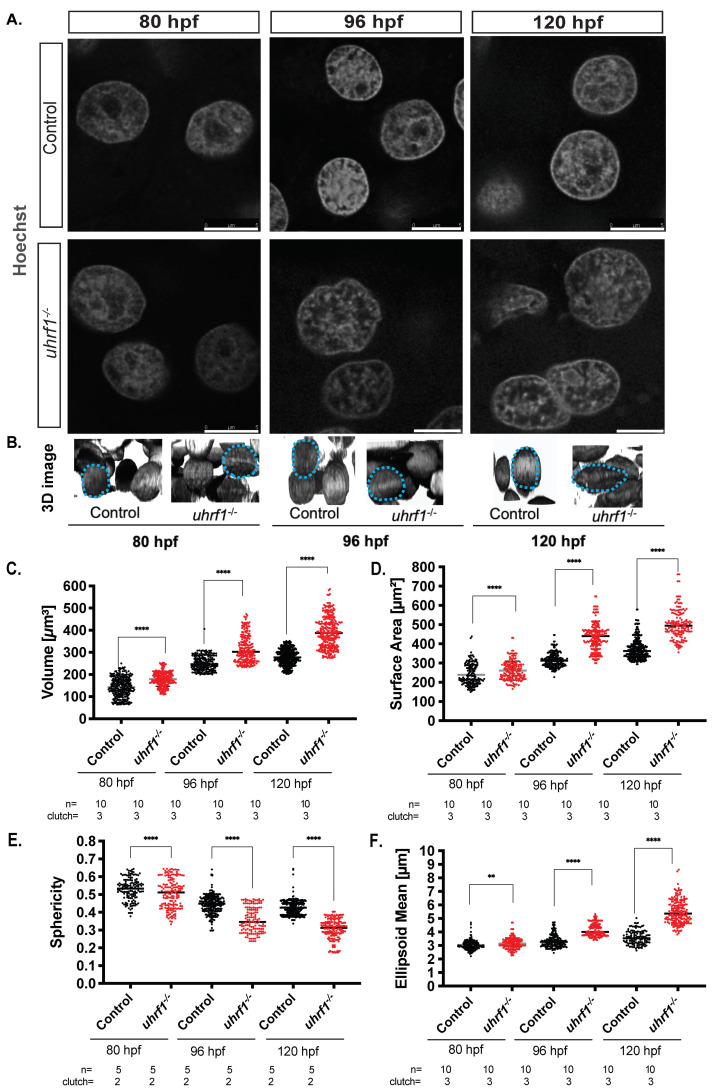
*uhrf1* loss increases hepatocyte nuclear size, disrupts nuclear morphology, and increases nucleoli number at 120 hpf. (**A**). Confocal images of phenotypically WT siblings and *uhrf1^−/−^* nuclei stained by Hoechst (in grey). (**B**). The 3D reconstructed nuclei in the liver of 80, 96, and 120 hpf WT siblings and *uhrf1^−/−^* nuclei larvae display changes in size and shape at all time points between controls and mutants. (**C**–**F**). Nuclear morphological measurements of control and *uhrf1^−/−^* hepatocyte nuclei scored at 80, 96, and 120 hpf for volume (**C**), surface area (**D**) for size measurements, and sphericity (**E**), ellipsoid mean (**F**) for shape measurements. Each dot represents one cell. All analyses were performed on 2 or 3 clutches, with minimum 2 livers analyzed per clutch. ** *p* < 0.005, **** *p* < 0.00005. Magnification: 63×, 4× zoom, Scale bar: 5 μm.

**Figure 6 genes-12-01081-f006:**
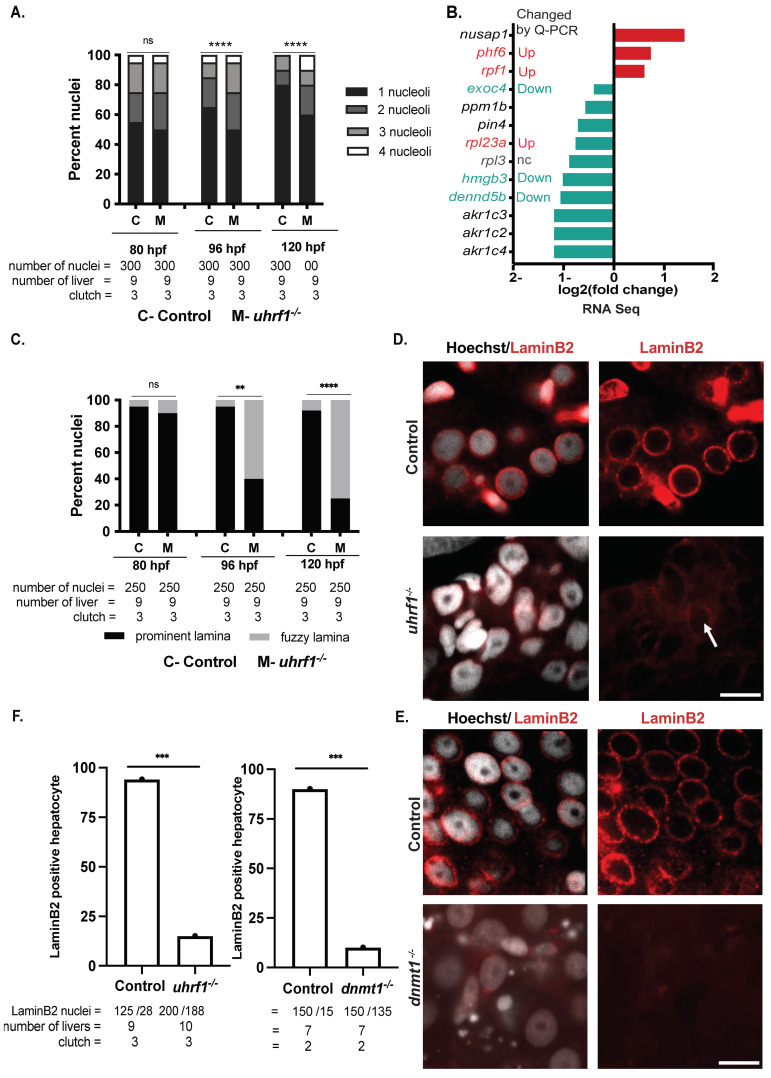
*uhrf1* loss disrupts hepatocyte nucleoli and laminB2 organization. (**A**). Control and *uhrf1^−/−^* hepatocyte nuclei scored at 80, 96, and 120 hpf for the number of nucleoli per hepatocyte. (**B**). Differential expression of nucleoli-associated genes in 120 hpf *uhrf1^−/−^* livers, nc: not significantly changed (**C**). Control and *uhrf1^−/−^* hepatocyte nuclei scored at 80, 96, and 120 hpf for the presence of normal lamina. (**D**,**E**). Confocal images comparing the immunostaining of LaminB2 in 120 hpf *uhrf1^−/−^* and *dnmt1^−/−^* livers with their respective WT controls. (**F**). Quantitative changes in LaminB2 staining in 120 hpf *uhrf1^−/−^* and *dnmt1^−/−^* hepatocyte nuclei. The numbers of LaminB2 positive nuclei are indicated below each bar. White arrows mark LaminB2 positive hepatocyte nuclei. All analyses were performed on 2 or 3 clutches, with minimum 3 livers analyzed per clutch. ** *p* < 0.005, *** *p* < 0.0005, **** *p* < 0.00005. Magnification: 63×, Scale bar: 7 μm.

## Data Availability

The datasets presented in this study can be found in GEO: ATAC-Seq: GEO (GSE173792), RNA-Seq: GEO (GSE130800 and GSE104953), RRBS-Seq: GEO (GSE173792).

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
