# Peer review of "Nuclear Organization during Hepatogenesis in Zebrafish Requires Uhrf1"

_genes, 2021, doi:10.3390/genes12071081_

Round 1

Reviewer 1 Report

This manuscript focuses on investigating how chromatin accessibility and DNA methylation levels influence the nuclear organization during hepatocyte differentiation in zebrafish. Using confocal microscopy and the quantification in the Tg line, the authors identified distinct nuclear organization and heterochromatin formation in mature hepatocytes. Authors performed ATAC-seq in hpf 120 liver cells to identify open chromatin regions, showing enrichment of developmental pathways in genes near the peaks and TE genes. Methylome analysis at 120 hpf showed enrichments of DNA methylation in specific TE classes, and the integration of methylome with ATAC-seq data discovered an increase in DNA methylation in exons outside of ATAC-seq peaks compared to one in exons inside of ATAC-seq peaks. The authors also performed transcriptome analysis at 78 and 120 hpf, revealing genes expressed at stage specifically and commonly. Finally, the DNA hypomethylation mutant, uhrf1, displayed distinct hepatocyte nuclear morphology, including loss of nuclear lamin, which was also shown in dnmt1 mutants. The authors concluded that the phenotype shown in uhrf1 is due to a loss of DNA methylation.  

Major issues
1)  Line 357: only 5% of the detected open chromatin covering promoters is very low compared to the general features of open chromatins shown in other studies. The authors should explain potential reasons why it is so low including technical issues. Since peaks from three replicates were combined, there is no information on how many of these peaks were reproducible. 
2) Promoter definition varied: in the ATAC-seq analysis, the promoter was defined as +/- 2000 bp from the TSS while it was defined as +/ 1000 bp in the methylome analysis. Any specific reason for the inconsistency? Data would likely look different with a consistent parameter of either +/-2000 or 1000 bp.  
3) In RNA-seq analysis, the authors performed a Venn diagram analysis and k-means clustering. However, the Venn diagram gave us the information of whether they were expressed or not. What authors wanted to know is what genes were differentially expressed between 78 and 120 hpf using tools such as DESeq2. Similarly, the k-means is a tool in which genes are forced to be categorized into the given number of k. It does not perform the statistic modeling for differential expression. In addition, in this study (Fig 4B), the k-mean clustering does not provide much information because the replicates (I assume the column in the heatmap is a replicate) showed a high level of variation, which likely influenced the clustering. 

Minor issues
1) Line 41-43 and 101-108, add references
2) Line 424-426, intron data needs to be mentioned. 
3) Line 594-596, the inverse correlation between DNA methylation and chromatin accessibility is valid only for exons. The authors need to tone it down.

Reviewer 2 Report

In this study, the authors tested the hypothesis that hepatocyte maturation during development was characterized by the acquisition of distinct nuclear morphology and a chromatin landscape by the pattern of DNA methylation in hepatocytes. They found out that uhrf1 mutant hepatocytes maintain a high level of nucleoli during hepatocyte maturation and are likely directly linked to a loss of DNA methylation. While of interest, the new findings of this manuscript are very limited, as there have been other studies on liver growth, development, and its relationship with the cell cycle regulator uhrf1. Below are some comments to help polish this manuscript.

Comments

  1. Page 3, line 107: What is meant by (ref)? Secondly, that sentence needs to be referenced.
  2. Page 4, line 152: The sentence sounds as if the following procedure was DNA extraction, and makes it a bit confusing to readers. Please revise.
  3. Was there any technical replicate to demonstrate the variability of the protocols used in this experiment?
  4. Why was ATAC-seq considered in this experiment over other genome accessibility or profiling technique? ATAC-seq is known to require high sequencing coverage to accurately map factors and has poor repeatability.
  5. Page 13, line 471: The word ‘expressed’ is having a different font style, it should be corrected.
  6. Page 17: Figures D and F are not labeled.
  7. Page 20, Line 614: The abbreviation E11.5 should be written in full.
  8. Page 22, line 695: For the sentence ‘… it is possible that UHRF1 in cancer acts as a …’ a reference should be cited.
  9. What is the essence and correlation between the mouse studies in the discussion and this current study?

Reviewer 3 Report

Comments to the Author

In this manuscript, the authors investigate nuclear organization during hepatogenesis requires Uhrf in the zebrafish model. Authors showed, ATAC-Seq analysis along with DNA methylation profiling of zebrafish livers at 120 hpf showed that closed, highly methylated chromatin occupies most transposable elements and that open chromatin correlated with gene expression. Further, DNA hypomethylation due to mutation of genes encoding ubiquitin-like, containing PHD and RING Finger Domains 1 (uhrf1) and DNA methyltransferase (dnmt1) did not block hepatocyte differentiation. Mutation in uhrf1 and dnmt1 had dramatic effects on the nuclear organization such as, large, deformed nuclei with multiple nucleoli, downregulation of nucleolar genes, and a complete lack of the nuclear lamina (lamin B2 staining) in uhrf1 mutation. This was phenocopied by dnmt1 mutation. The authors concluded hepatocyte nuclear morphogenesis coincides with organ morphogenesis and outgrowth and that DNA methylation directs chromatin organization and in turn nuclear shape and size during liver development. The results are of interest; overall, the manuscript was well written. However, some suggestions and concerns must be addressed.

Major comments

  1. ATAC-Seq : did authors used only hepatocytes or total liver cell population. Line 332: GFP expression was used to verify hepatocyte identity. Did authors cell sorted for GFP +ve cells before ATAC-Seq? Since, liver is populated with other cells for example hepatic macrophages (PMID: 26154973 & PMID: 31867007).
  2. RNA-Seq of 120 hpf uhrf1 mutant livers showed changes in nucleoli-associated genes; whether authors validated these RNA-Seq results with qPCR? It is ideal to include qPCR for fewer genes to confirm the RNA-Seq.
  3. Lamin B2 staining methodology not provided in the manuscript must be included. Moreover, what is the rationale for choosing Lamin B2 instead of other Lamin proteins?
  4. Did the authors observe any phenotypes related to Laminopathies or envelopathies in Dnmt1 and Uhrf1?

Minor Comments

  1. Must be included wherever applicable the following items catalog numbers, company, state, and country for all chemicals, antibodies, and kits.
  2. Provide scale bar values and magnifications in the legend of the microscopic images.
  3. Line 107… occur simultaneously (ref) – Reference missing.
  4. Do avoid repetition of methods again in result sections, wherever applicable (For example Section 2.4 for ATAC-Seq methodology and results from 3.2)
  5. Say, for example, there were studies showed Uhrf1 and Dnmt1role on other organs may be included (PMID: 33170271, PMID: 21126517, PMID: 31394080, and others), if relevant.
  6. Over-statement must be avoided throughout the manuscript– abstract and conclusion sections. Do avoid overstatements throughout the manuscript.
  7. There are many other minor errors of syntax and grammar throughout the text, which need to be fixed.

Round 2

Reviewer 1 Report

The authors' responses and changes in the main text have addressed my main concerns.